# A Double Mutation in the *ALS* Gene Confers a High Level of Resistance to Mesosulfuron-Methyl in Shepherd’s-Purse

**DOI:** 10.3390/plants12142730

**Published:** 2023-07-23

**Authors:** Huan Lu, Yingze Liu, Dexiao Bu, Fan Yang, Zheng Zhang, Sheng Qiang

**Affiliations:** Weed Research Laboratory, College of Life Sciences, Nanjing Agricultural University, Nanjing 210095, China; huan.lu@njau.edu.cn (H.L.); 2020116013@stu.njau.edu.cn (Y.L.); budexiaoa@163.com (D.B.); 2022116010@stu.njau.edu.cn (F.Y.); zhangzheng@njau.edu.cn (Z.Z.)

**Keywords:** ALS-inhibiting herbicide, double mutation, herbicide resistance, target-site resistance, shepherd’s-purse

## Abstract

Shepherd’s-purse (*Capsella bursa-pastoris*), a globally distributed noxious weed species often found in wheat, has evolved resistance to ALS-inhibiting herbicides mainly due to single mutations in the *ALS* gene. In the present study, dose–response bioassays showed that a shepherd’s-purse population (R), collected from Xinghua, Jiangsu Province, China, had high level of resistance to the ALS-inhibiting herbicide, mesosulfuron-methyl (800-fold), and even much higher resistance levels to other reported ALS-inhibiting herbicides, tribenuron-methyl (1313-fold), bensulfuron-methyl (969-fold) and penoxsulam (613-fold). Sequencing of the open reading frame of the *ALS* gene revealed a double *ALS* gene mutation (Pro197-Ser plus Trp574-Leu) conferring the high resistance in the R plants. Docking analysis of the ALS protein and mesosulfuron-methyl predicts that the two amino acid substitutions in the R samples reduces the binding energy to the herbicide by decreasing the hydrogen bonds (H-bonds) and other interactions, thus endowing resistance to ALS-inhibiting herbicides. These results demonstrate that the double ALS mutation confers high resistance levels to ALS-inhibiting herbicides. To our knowledge, this is the first evidence of the double ALS mutation in shepherd’s-purse endowing ALS-inhibiting herbicide resistance.

## 1. Introduction

Acetolactate synthase (ALS)-inhibiting herbicides, also known as acetohydroxyacid synthase (AHAS)-inhibiting herbicides, have been commercialized since the 1980s. The herbicides have five main chemical classes, sulfonylurea (SU), imidazolinones (IMI), triazolopyrimidine (TP), pyrimidinyl thiobenzoate (PTB) and sulfonylamino-carbonyl-triazolinones (SCT), according to distinct structures [1]. ALS-inhibiting herbicides inhibit the first enzyme in the biosynthesis of the branched-chain amino acids (Val, Leu and Ile), leading to depletion of branched-chain amino acids and eventually plant death [2]. Due to the broad spectrum weed control, low mammalian toxicity and selectivity in major crops, they have been widely used worldwide. However, the intensive application has inevitably resulted in the evolution of herbicide resistance in weedy species. So far, 171 weedy species have evolved resistance to ALS-inhibiting herbicides [3].

Weeds resist herbicides mainly through two different mechanisms, target-site resistance (TSR) and non-target-site resistance (NTSR) mechanisms. TSR primarily refers to mutations of target proteins; thus, reducing or preventing herbicide binding. The 30 substitutions at eight known amino acid residues have been documented to confer resistance to ALS-inhibiting herbicides, including Ala122, Pro197, Ala205, Asp376, Arg377, Trp574, Ser653 and Gly654 [3,4]. In addition, overexpression of target protein endows target-site-based resistance to ALS-inhibiting herbicides and other herbicide modes of action (MOAs) [5,6]. NTSR refers to mechanism(s) reducing the lethal herbicide dose reaching the target site in plants, mainly including reduced herbicide uptake and/or translocation, and enhanced herbicide metabolism. NTSR to ALS-inhibiting herbicides has been reported in many weeds, e.g., *Lolium rigidum* and *Echinochloa* spp. and several cytochrome P450 genes (*CYP81A12*, *CYP81A21* and *CYP81A10v7*) have been characterized [7,8,9,10,11,12].

Shepherd’s-purse (*Capsella bursa-pastoris*), as a self-compatible tetraploid dicot species, widely infests wheat (*Triticum aestivum*) fields [13]. This weed severely affects wheat production due to competitive advantages and prolific production of seeds [14]. Although ALS-inhibiting herbicides have effectively controlled shepherd’s-purse for three decades, evolution of resistance to ALS-inhibiting herbicides has been reported in shepherd’s-purse since the 2000s. So far, target-site base ALS mutation is the main mechanism for shepherd’s-purse resistance to ALS-inhibiting herbicides, focusing on the substitutions at the two separate amino acid residues, Pro197 and Trp574 [15,16,17,18,19]. In this study, a putative mesosulfuron-methyl-resistant shepherd’s-purse population was collected from Xinghua, Jiangsu, China. The objectives of this study were to (1) confirm the resistance level to mesosulfuron-methyl; (2) characterize the resistance patterns to herbicides with different MOAs; and (3) elucidate the resistance mechanism(s) in the population.

## 2. Results

### 2.1. Mesosulfuron-Methyl Dose-Response in S and R Shepherd’s-Purse Populations

As expected, the S shepherd’s-purse plants were sensitive to mesosulfuron-methyl, showing approximately 40% mortality at 1.0 g ha^−1^ (1/16-fold of field rate). The S plants were all dead with the 1/2-fold field rate of mesosulfuron-methyl. Whereas, the R plants exhibited more than 90% survival even at 252.8 g ha^−1^ (16-fold of field rate) (Figure 1). The LD50 values of the S and R populations were estimated as 0.8 and 640 g ha^−1^, respectively (Table 1), revealing that the R population is 800-fold more resistant to mesosulfuron-methyl than the S population.

### 2.2. Cross-Resistance Patterns to Other ALS-Inhibiting Herbicides

The S and R plants were further treated with three other ALS-inhibiting herbicides, tribenuron-methyl, bensulfuron-methyl and penoxsulam. The results showed that all S plants died at half of the field rates (for all three herbicides), while the R plants survived all at the rates of 16-fold of the field rates. Comparisons of R/S LD_50_ values revealed that the R population was 1313-, 969- and 613-fold more resistant to tribenuron-methyl, bensulfuron-methyl and penoxsulam, respectively, relative to the S population (Table 1). In addition, the resistance patterns to other herbicide MOAs were tested. The results showed that the R population did not have resistance to 2,4-D (2-(2,4-dichlorophenoxy)acetic acid, active ingredient 30%), atrazine (6-chloro-4-N-ethyl-2-N-propan-2-yl-1,3,5-triazine-2,4-diamine, active ingredient 38%), diflufenican (N-(2,4-difluorophenyl)-2-[3-(trifluoromethyl)phenoxy]pyridine-3-carboxamide, active ingredient 41%) or mesotrione (2-(4-methylsulfonyl-2-nitrobenzoyl)cyclohexane-1,3-dione, active ingredient 40%).

### 2.3. Sequencing of ALS Genes in Both S and R Plants

Shepherd’s-purse is a tetraploid weedy species and has two *ALS* gene copies [18]. A pair of primers (mentioned in the Section 4.3) was used for ORF of *ALS* gene cloning in both S and R plants. Two *ALS* gene copies, showing 98% identity, were cloned from both S and R plants (Appendix A). Amino acid alignment of ALS proteins from the S and R plants revealed two amino acid substitutions at the position of Pro197 and Trp574, leading to the widely known Pro197-Ser and Trp574-Leu mutations, respectively (Table 2, Appendix A). Sequencing comparisons revealed that both *ALS* gene copies only contained the two mutations simultaneously and sequencing results showed only one peak without noises at the tested positions in the R samples (Figure 2, Appendix A), indicating that the R population has a double *ALS* gene mutation (Pro197-Ser + Trp574-Leu) in the two *ALS* gene copies; thus, endowing a high level of resistance to mesosulfuron-methyl.

### 2.4. Docking of ALS Protein and Mesosulfuron-Methyl

Two ALS protein isoforms (I and II) (Appendix A), showing 98% identity, were obtained, similar to the results reported by Wang, et al. [18]. The three-dimensional (3D) structures of ALS protein isoform I of both S and R plants were constructed. The docking results showed that mesosulfuron-methyl binds to amino acid residue Pro197, Met200, Phe206, Lys256, Gln260, His352, Arg377 and Trp574 of ALS protein in the S (wild type, WT), and the herbicide binds to Gln207, Lys256, Asp376, Arg377 and Ser653 in the R (mutant) (Figure 3). Mesosulfuron-methyl forms five hydrogen bonds (H-bonds) in the WT complex, three with Arg377, one with Lys256 and one with Trp574, while it only forms two H-bonds with Gln207 and Lys256 in the mutant complex. Except for H-bonds, other interactions were also observed in two docking results. In the WT complex, interactions of π-π, π-Alkyl and π-Sulfur were shown between mesosulfuron-methyl and the protein. However, obviously fewer interactions were shown in the mutant complex compared to that in the WT complex (Figure 4). The interaction energy of mesosulfuron-methyl to the WT was −58.66 kcal/mol, and the energy to the mutant was −42.79 kcal/mol (Appendix A), indicating that mesosulfuron-methyl had a higher binding affinity (lower interaction energy) for ALS binding in the S than in the R plants. Docking between mesosulfuron-methyl and ALS protein isoform II of S and R plants displayed similar results (Appendix A, Appendix A), exhibiting less H-bonds and higher interaction energy in the mutant.

## 3. Discussion

In this study, a shepherd’s-purse population was confirmed to be highly resistant to ALS-inhibiting herbicide mesosulfuron-methyl due to a double *ALS* gene mutation. Further molecular docking analysis revealed that reduction of interaction between the ALS protein mutant and the herbicide could be the main reason for the resistance in the R plants.

Tribenuron-methyl has been the primary herbicide used to control shepherd’s-purse during the past decades. Previous studies have, thus, focused on shepherd’s-purse resistance evolution to tribenuron-methyl [15,17,18,19]. These reported resistant shepherd’s-purse populations had an over 15-year use history of tribenuron-methyl in China [19]. Whereas the wheat field where the R population was collected, in contrast to the previous cases, had a 5-year mesosulfuron-methyl use history. Compared to over 15-year use history, the 5-year use history, a relatively short period, has still led to resistance to ALS inhibitors in shepherd’s-purse. This indicates that herbicide resistance has evolved rapidly in this shepherd’s-purse population. Rapid evolution of resistance to mesosulfuron-methyl has also been reported in populations of many weedy species, e.g., *Aegilops tauschii* [20], *Alopecurus aequalis* [21], *Beckmannia syzigachne* [22], and *L. rigidum* [23]. These weed species usually had a 3–5-year consecutive use history of ALS inhibitors or other herbicide MOAs, resulting in the resistance conferred by TSR or NTSR.

Sequencing of the *ALS* gene in both S and R shepherd’s-purse plants revealed that a double mutation (Pro197-Ser plus Trp574-Leu) identified in the R plants is responsible for the high resistance levels to ALS-inhibiting herbicides. Sequencing results revealed that all test R samples had the two mutations, and both mutated sites had only a single signal peak without noise (Figure 2 and Table 2), indicating that the *ALS* gene double mutation could be homozygous. Substitutions at amino acid residues Pro197 and Trp574 have been widely known to confer ALS-inhibiting herbicide resistance. Thus far, 12 different amino acid substitutions (His, Thr, Arg, Leu, Gln, Ser, Ala, Ile, Met, Lys, Glu, and Tyr) at Pro197 have been reported to confer the resistance [24]. The substitutions at the Pro197 residue confer resistance to SU, but have distinct resistance profiles to other ALS inhibitor chemical groups, depending on amino acid substitutions and species. The shepherd’s-purse population with the Pro197-Ser mutation showed a high resistance level (200- to 300-fold) to SU herbicide tribenuron-methyl compared to the S population based on the GR_50_ R/S ratios, but were susceptible to IMI and TP herbicides [15]. In addition, the amino acid substitutions (Leu, Met, Arg and Gly) at the Trp574 residue always confer broad spectrum resistance to SU, IMI and TP, as the residue Trp574 is vital for the shape of active-site channel of ALS protein [1,25,26]. The shepherd’s-purse population with the Trp574-Leu mutation had a more than 300-fold resistance level to tribenuron-methyl [19]. A concurrence of Pro197-Leu and Trp574-Leu was confirmed in a shepherd’s-purse population from Henan Province, China, endowing a high level of resistance (600-fold) to tribenuron-methyl [19]. However, the authors did not mention whether the two mutations occurred in a same *ALS* gene copy. Shepherd’s-purse has two different *ALS* gene copies [18]; the concurrence of Pro197-Leu and Trp574-Leu could occur at two copies separately, which is not a double mutation. Our results showed a double mutation (Pro197-Ser plus Trp574-Leu) occurs in both *ALS* gene copies in the R shepherd’s-purse population, conferring high levels of resistance (600- to 1300-fold based on the R/S LD_50_ ratios) to SU (mesosulfuron-methyl, tribenuron-methyl and bensulfuron-methyl) and TP (penoxsulam). The herbicide doses used in the S population in our study were comparable to the rates used for other susceptible populations in the references [15,18,19], indicating that the S population likely shares a similar sensitivity pattern to ALS-inhibiting herbicides, relative to the susceptible populations in the references. The double ALS mutation reported in the study endows obviously higher resistance levels to ALS-inhibiting herbicides than the single ALS mutations; even higher than the concurrence of Pro197-Leu and Trp574-Leu mentioned before. To our knowledge, this is the first report on a double *ALS* gene mutation conferring cross-resistance to ALS-inhibiting herbicides. Double target gene mutation conferring herbicide resistance has been rarely reported in weed species. The most typical example of double target gene mutation is the Thr102-Ile + Pro106-Ser (TIPS) of *EPSPS* (5-Enolpyruvylshikimate-3-Phosphate Synthase) gene in *Eleusine indica* [27]. The *E. indica* population containing TIPS showed a very high level of resistance to glyphosate (>182-fold), compared to the single mutation, Pro106-Ser, (5.6-fold). Double mutation usually confers a high level of herbicide resistance, which is vital for the survival of weeds under herbicide stress. However, the high level of herbicide resistance conferred by the double target gene mutation also brings great difficulties for weed control, because the normal herbicide field rate cannot effectively kill the weeds. Once the resistant weeds, especially containing double target gene mutations, occur, herbicide resistance can be evolved rapidly under continuous herbicide applications. This is a big threat for field weed control and crop production.

Studies have already identified fitness cost associated with resistance to ALS-inhibiting herbicides in a weed species, *Alopecurus aequalis* [28]. Pro197-Tyr and Trp574-Leu mutations cause a decrease and increase in growth in *A. aequails*, respectively. Thus, the plants containing a Trp574-Leu mutation is possibly more persistent than plants containing a Pro197-Tyr mutation, after herbicide application is discontinued. *E. indica* plants containing TIPS, mentioned above, exhibited a significant fitness cost, especially in biomass, seed number and inflorescence mass [29]. While a survey on fitness of different ALS mutations in wild radish (*Raphanus raphanistrum*) revealed that, although the *ALS* gene mutations changed ALS enzyme activities differently, they did not impose negative effects on plant growth and resource-competitive ability [30]. These results indicate that ALS target-site mutations did not affect fitness in wild radish, and, thus, contributing to high frequencies of ALS mutations and its fast evolution of resistance to ALS inhibitors. Target-site gene mutations cause distinct fitness patterns in different weed species. These fitness patterns might be related to herbicide resistance–fitness trade off in weed species. In our study, the double mutation may endow a certain level of fitness cost because the R plants exhibited growth reduction at the early growth stage in comparison with the S plants, which is worthy of further study.

So far, crystal structures of ALS protein complex with 13 different herbicides in Arabidopsis (*Arabidopsis thaliana*) have been determined [31], providing an exquisite understanding on bindings of herbicides to the protein. ALS-inhibiting herbicides bind across the binding pocket to block substrate access [32], for which the binding domain is formed by 18 amino acid residues [25]. Mesosulfuron-methyl, as an SU herbicide, binds the conserved binding domain with different orientations. Our results exhibited that mesosulfuron-methyl binds to the conserved amino acid residues (e.g., Pro197, Arg377, Trp574), similar to the previous study [33]. Mesusulfuron-methyl forms five H-bonds to the residue Lys256, Arg377 and Trp574 of the WT (ALS protein of the S plants), while the H-bonds decreased to two in the binding between the herbicide and the mutant (ALS protein of the R plants) (Figure 3). Moreover, other interactions were also reduced in the mutant complex (e.g., interactions of π-π and π-Sulfur were vanished in the mutant complex, Figure 4). Trp574 of the ALS protein is an important residue for ligand binding. In addition to H-bond formation, mesosulfuron-methyl forms a π-π interaction with the Trp574 in the WT complex, similar to the chlorimuron-ethyl binding [33]. However, this π-π interaction was abolished in the mutant complex (Figure 4). Taken together, the binding of mesosulfuron-methyl to the mutant is less stable compared to that of the herbicide to the WT (Appendix A). The molecular docking analysis revealed that the weakened binding caused by the double *ALS* gene mutation is very likely responsible for mesosulfuron-methyl resistance in the R population.

Mesosulfuron-methyl is an indispensable herbicide in wheat fields for management of problematic weeds, such as *A. tauschii*, *A. myosuroides* and a few broad-leaf weeds. However, it has failed to control this resistant shepherd’s-purse population recently due to its high resistance conferred by the double *ALS* gene mutation. Significantly, the R population did not show any resistance to the tested herbicides with other MOAs, which is vital for the control of this population. Based on its herbicide use history, long-term sole use of the ALS-inhibiting herbicide mesosulfuron-methyl is likely one of the main reasons for the sensitivity to other herbicide MOAs. Our results suggest that herbicide alterations or an herbicide mix should be used to control the populations resistant to ALS-inhibiting herbicides, and, thus, suppress/delay herbicide resistance evolution in shepherd’s-purse.

## 4. Materials and Methods

### 4.1. Materials

Seedlings of shepherd’s-purse (*Capsella bursa-pastoris*) were collected from wheat belt in Xinghua, Jiangsu Province, China (119.883845° E, 33.080425° N). The collected seedlings were sprayed with mesosulfuron-methyl at the field rate (15.8 g ha^−1^) in the greenhouse and seeds were collected from the survivors (27 survivors, resistant to mesosulfuron-methyl, R) with regular watering and fertilizing. Another shepherd’s-purse population without any herbicide use history, collected from wasteland in Baima, Nanjing, Jiangsu Province (119.188889° E, 31.622222° N), was used as a susceptible population (S).

### 4.2. Dose-Responses to ALS-Inhibiting Herbicides

Seeds of the shepherd’s-purse S and R populations were germinated on moist filter paper at room temperature for 3–4 days. Germinated seeds were transplanted to pots (15-cm diameter) containing potting mix (50% peat moss, 25% vermiculite and 25% pine bark) with 10 seedlings per pot. When the seedlings reached the two- to three-leaf stage (about two-week growing after transplanting), a range of doses of different ALS-inhibiting herbicides (Table 3) were sprayed to both S and R plants. Herbicide treatments were conducted using an enclosed cabinet sprayer delivered 450 L ha^−1^ in two passes at 0.4 MPa. There were four replicates in each treatment for each dose–response. Treated plants were grown in a glasshouse with regular watering during normal growing season (November to February). Plant survival was assessed 21 days after treatment (DAT). In addition, 2,4-D (675 g ha^−1^, supplied by KingAgroot Co., Ltd., Qingdao, China), atrazine (1500 g ha^−1^, Jiangsu Fuding Chemical Co., Ltd., Nanjing, China), diflufenican (120 g ha^−1^, Shandong Sino-Agri United Biotechnology Co., Ltd., Weifang, China) and mesotrione (100 g ha^−1^, Jiangsu Agrochem Laboratory Co., Ltd., Changzhou, China) were also sprayed to identify the multiple resistance patterns of the R population.

### 4.3. Cloning and Sequencing of ALS Gene

DNA was extracted from the leaf tissue (two- to three-leaf stage) of untreated S plants and R surviving 15.8 g mesosulfuron-methyl ha^−1^ (10 plants per population) using the CTAB method [34]. The open reading frame (ORF) regions of the *ALS* gene from the S and R plants were cloned using the primer pair (F: 5′-GGTGCATCAATGGAGATTCA-3′, R: 5′-TCAGTATTTAGTCCGGCCATC-3′) designed based on the sequence of the shepherd’s-purse *ALS* gene (NCBI ID: HQ880660.1). Polymerase chain reaction (PCR) was conducted in a 25 μL volume, consisting of 2 μL DNA, 0.5 mmol L^−1^ of each primer and 12.5 μL of EmeraldAmp PCR Master Mix (Takara Bio Inc., Shiga, Japan), running with the following steps: 95 °C for 5 min, 35 cycles of 95 °C for 30 s, 55 °C for 30 s, and 72 °C for 60 s, followed by a final extension of 7 min at 72 °C. The PCR product was purified from agarose gel (1%) using a FastPure Gel DNA Extraction Mini Kit (Vazyme Co., Ltd., Nanjing, Jiangsu, China). The purified fragment was then cloned into the pMD19-T vector (Takara) and transformed to *Escherichia coli* (Takara). Five white colonies from each sample were selected for sequencing. Sequencing results were analyzed by DNAMAN 8.0 (Lynnon Corp., Quebec, QC, Canada).

### 4.4. Molecular Docking between ALS Protein and Mesosulfuron-Methyl

Two *ALS* gene copies were obtained from both S and R populations. The conserved region of ALS proteins of S and R shepherd’s-purse plants was constructed based on the template of Arabidopsis ALS protein (PDB code: 7tzz) using online modelling tool Swissmodel (https://swissmodel.expasy.org/interactive, accessed on 7 December 2022). Mesosulfuron-methyl, downloaded from PubChem (https://pubchem.ncbi.nlm.nih.gov, accessed on 7 December 2022), was selected as the ligand. The binding pocket was confirmed by Lonhienne, et al. [33]. Molecular docking between ALS protein and mesosulfuron-methyl was conducted by Biovia Discovery Studio 2019 (Dassault Systemes, Paris, France) with the CDocker method. The number of top poses was set to 10 and the radius of pose clustering was set to 0.5 Å, with the other parameters were kept as default for each docking. Molecular docking between ALS protein and mesosulfuron-methyl was individually optimized based on the energy minimization and molecular dynamics simulations.

### 4.5. Statistical Analysis

Herbicide dose causing 50% plant mortality (LD_50_) was estimated by non-linear regression analysis (a three-parameter Sigmoidal-logistic model) by SigmaPlot 13.0 (Systat Software, Inc. San Jose, CA, USA): y = *a*/[1 + (x/x_0_)*^b^*], where a is the upper asymptote, x_0_ equals the LD_50_ value, and *b* is the slope at x_0_. The LD_50_ values of the S and R populations were compared by *t*-test using Prism 9.0 (GraphPad Software, La Jolla, CA, USA).

## Figures and Tables

**Figure 1 plants-12-02730-f001:**
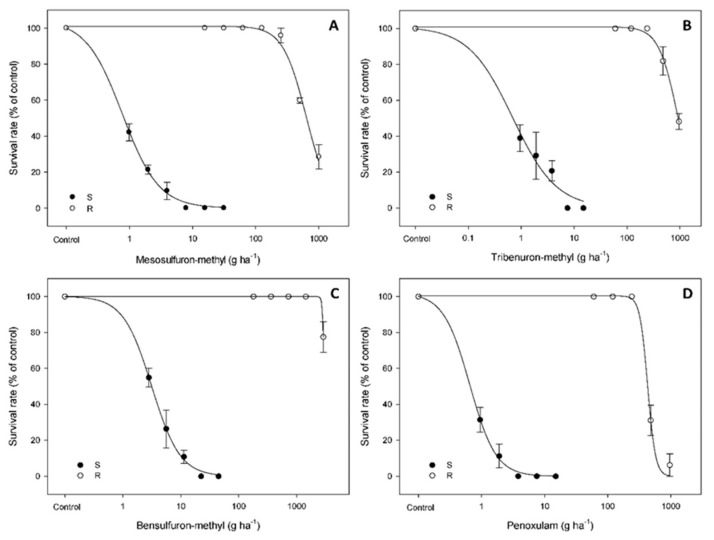
Percentage survival of the S and R shepherd’s-purse plants in response to four ALS-inhibiting herbicide treatments ((**A**), mesosulfufon-methyl; (**B**), tribenuron-methyl; (**C**), bensulfuron-methyl; (**D**), penoxsulam) estimated by a three-parameter Sigmoidal-logistic model.

**Figure 2 plants-12-02730-f002:**
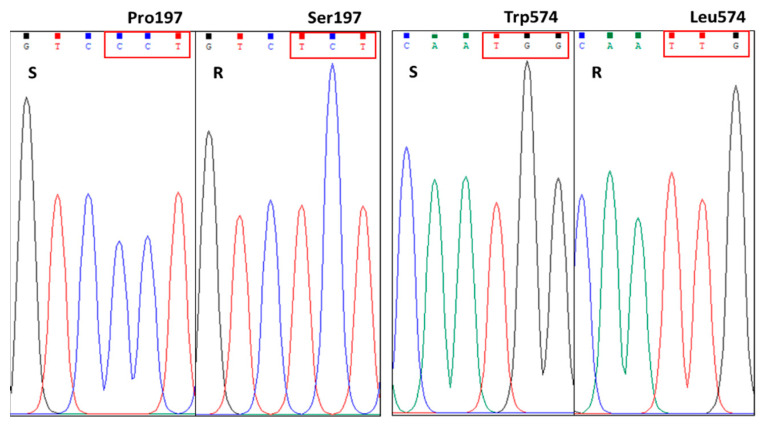
*ALS* gene sequencing of at the position of amino acid residue 197 and 574 in the S and R shepherd’s-purse plants.

**Figure 3 plants-12-02730-f003:**
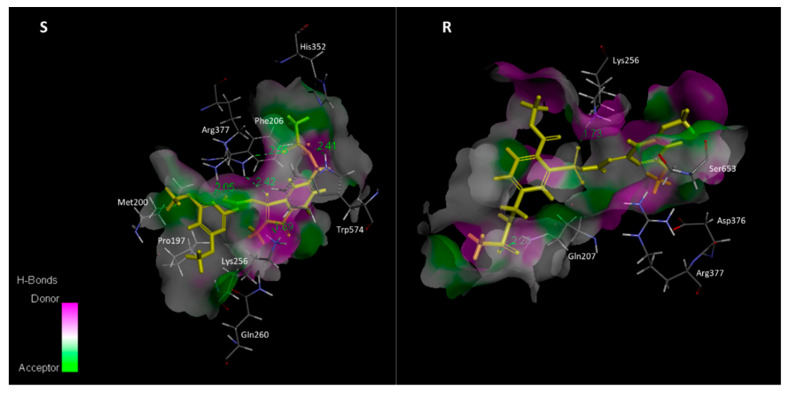
Spatial structure of contact interface of mesosulfuron-methyl (yellow) in S (WT) and R (Pro197-Ser + Trp574-Leu) shepherd’s-purse ALS protein isoform I (the protein contact surface is colored by H-bond donor or acceptor distribution; binding site amino acids are represented by sticks; intermolecular contacts are indicated by dashed lines, and the numbers are H-bond distance, Å. Amino acid numbering refers to Arabidopsis ALS protein).

**Figure 4 plants-12-02730-f004:**
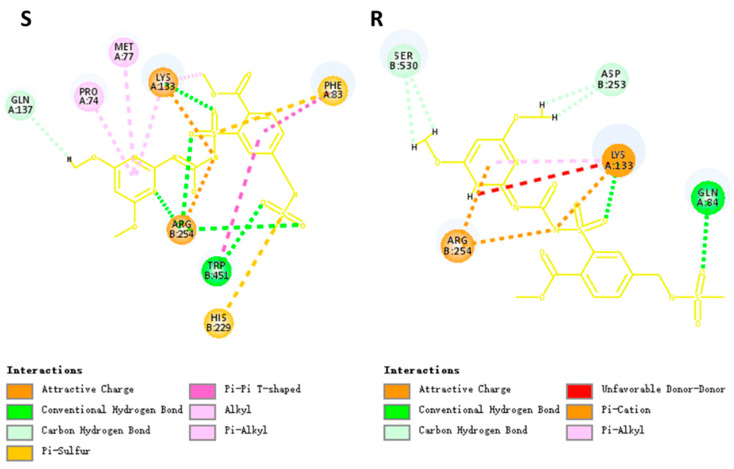
Two-dimensional diagram (2D-diagram) of interactions between mesosulfuron-methyl (yellow) and amino acid residues in S (WT) and R (Pro197-Ser + Trp574-Leu) shepherd’s-purse ALS protein (Amino acid residue Pro74, Met77, Phe83, Gln84, Lys133, Gln137, His229, Asp253, Arg254, Trp451 and Ser530 refers to Pro197, Met200, Phe206, Gln207, Lys256, Gln260, His352, Asp376, Arg377, Trp574 and Ser653 in Arabidopsis ALS protein, respectively).

**Table 1 plants-12-02730-t001:** The LD_50_ values and the relevant parameter estimates of dose–responses to four ALS-inhibiting herbicides using a three-parameter Sigmoidal-logistic model.

	*a*	*b*	R^2^	LD_50_ (g ha^−1^)	R/S Ratio
Mesosulfuron-methyl					
S	100 (1.8)	1.5 (0.1)	1.00	0.8 (0.1)	-
R	101 (1.6)	2.3 (0.3)	0.99	640 *** (31)	800
Tribenuron-methyl					
S	99.8 (6.6)	1.0 (0.3)	0.99	0.7 (0.2)	-
R	101 (1.2)	2.4 (0.2)	1.00	919 *** (30)	1313
Bensulfuron-methyl					
S	100 (2.0)	1.8 (0.1)	1.00	3.1 (0.1)	-
R	100 (0.0)	29 (2.3)	1.00	3005 *** (10)	969
Penoxsulam					
S	100 (1.7)	2.1 (0.3)	1.00	0.7 (0.0)	-
R	100 (2.0)	7.1 (3.5)	1.00	429 *** (26)	613

*** Asterisk represents the LD_50_ value of the R is significantly higher than that of the S population at the level of *p* < 0.001.

**Table 2 plants-12-02730-t002:** Polymorphisms of nucleotide and amino acid in *ALS* gene sequences from the S and R shepherd’s-purse plants.

Population	Polymorphisms			
	Nucleotide 589–591 *	Amino Acid 197	Nucleotide 1720–1722	Amino Acid 574
S	CCT	Pro	TGG	Trp
R	TCT	Ser	TTG	Leu

* Nucleotide and amino acid numbering refer to the Arabidopsis (*Arabidopsis thaliana*) *ALS* gene. The mutated nucleotides and the amino acid mutants were underlined and bold.

**Table 3 plants-12-02730-t003:** ALS-inhibiting herbicides used for dose-response study.

Herbicide	Population	Rate Used (g ha^−1^)	IUPAC Chemical Name	Active Ingredient	Manufacture
Mesosulfuron-methyl	S	0, 1, 2, 4, 8, 16, 32	methyl 2-[(4,6-dimethoxypyrimidin-2-yl)carbamoylsulfamoyl]-4-(methanesulfonamidomethyl)benzoate	3%	Bayer China Co., Ltd., Beijing, China
	R	0, 16, 32, 64, 128, 256, 512, 1024			
Tribenuron-methyl	S	0, 0.95, 1.9, 3.8, 7.5, 15	methyl 2-[[(4-methoxy-6-methyl-1,3,5-triazin-2-yl)-methylcarbamoyl]sulfamoyl]benzoate	10%	Shandong Vicome Greenland Chemical Co., Ltd., Jinan, China
	R	0, 60, 120, 240, 480, 960			
Bensulfuron-methyl	S	0, 2.8, 5.6, 11.3, 22.5, 45	methyl 2-[(4,6-dimethoxypyrimidin-2-yl)carbamoylsulfamoylmethyl]benzoate	30%	Jiangsu Institute of Ecomones Co., Ltd., Changzhou, China
	R	0, 180, 360, 720, 1440, 2880			
Penoxsulam	S	0, 0.95, 1.9, 3.8, 7.5, 15	2-(2,2-difluoroethoxy)-N-(5,8-dimethoxy-[1,2,4]triazolo[1,5-c]pyrimidin-2-yl)-6-(trifluoromethyl)benzenesulfonamide	2.5%	Corteva Agriscience Co., Ltd., Shanghai, China
	R	0, 60, 120, 240, 480, 960			

IUPAC: International Union of Pure and Applied Chemistry.

## Data Availability

Data will be made available on request.

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
