# Peer review of "A Double Mutation in the ALS Gene Confers a High Level of Resistance to Mesosulfuron-Methyl in Shepherd’s-Purse"

_plants, 2023, doi:10.3390/plants12142730_

Round 1
Reviewer 1 Report
The study is dedicated to a topical problem in plant protection, resistance of arable weeds to herbicides. The authors have found that a double single-point mutation confers resistance to ALS inhibitors to Capsella bursa-pastoris. The authors further explore the mechanism of herbicide resistance caused by this mutation.
The importance of this finding and the consequences (including further research questions) should be better explained in the Discussion part.
I do not quite understand the placing of figures and tables in the manuscript. Each figure/table should be placed as close as possible to the text where it is first mentioned.
Overall, the quality of the text is good but some minor changes are required.
Please also see some comments to the text below.
Ln 7 “notorious” - noxious? “global wheat fields” - probably, “a globally distributed weed species often found in wheat” or a similar text would be better.
Ln 14 molecular docking – or docking analysis?
Ln 25-25 and 31 suggest to use "herbicides" (plural)
Ln 33 “intensive applications” - suggest “application” (singular)
Ln 36 “two different mechanisms” - later in the text different mechanisms of the NTSR are mentioned, so these are not just two mechanisms.
Ln 41 Overexpression of the enzyme could rather be related to the NTSR, besides, the cited papers (references 5 and 6) do not mention this mechanism in relation to ALS inhibitors.
Ln 53 suggest “main” instead of “primarily”
Ln 67 could you please explain how the number 790 is derived?
Ln 70 replace “another three” with “three other”
Ln 85 a pair of primers “mentioned in section 2.3” - in the section 4.3
Figure 2 and Table 3 show the same information, perhaps one of these is sufficient
Ln 148 “molecular docking analysis” revealed (…)?
Ln 154 replace “different from” with “in contrast to” or similar
Ln 164 known to cause (not “known for”)
Ln 178 replace “that if the” with “whether the” (two mutations occurred)
Ln 180 “both two” - just “both" is sufficient
All scientific names of plants/microorganisms should be written in italic.
Ln 214 shouldn’t it read: studies identified fitness cost associated with resistance to ALS inhibitors?
Ln 217 worth of (further study)
Ln 220 “this gave us a hint” is stylistically inappropriate, replace with “this suggests that”
Ln 235 were transplanted?
Ln 237 when the seedlings reached … stage
Some minor correction are required
Reviewer 2 Report
It might be of interest to include (if you have any) images of the S and R populations of Shepherd's purse before and after application of herbicide - to show what the susceptible and resistant plants look like.
I was looking for more discussion on the significance of your results, especially since this is the first report of a double-mutation in the ALS gene. Can you discuss the significance of this double-mutation in terms of the evolution of herbicide resistance in this weed, and its implications in weed management and crop production? Also, can you add to your discussion the rate or speed of the evolution of herbicide resistance in other weeds compared to shepherd's purse?

Some of the sentences can be revised for more clarity or better flow for readers. I have highlighted these sentences.
